



# Pattern scaling of simulated vegetation change in North Africa during glacial cycles

Mateo Duque-Villegas[1,2], Martin Claussen[1,3], Thomas Kleinen[1], Jürgen Bader[4], and Christian H. Reick[1]

[1]Max Planck Institute for Meteorology, Hamburg, Germany
[2]International Max Planck Research School on Earth System Modelling, Hamburg, Germany
[3]Meteorological Institute, Center for Earth System Research and Sustainability (CEN), Universität Hamburg, Hamburg, Germany
[4]Institute of Oceanography, Center for Earth System Research and Sustainability (CEN), Universität Hamburg, Hamburg, Germany

**Correspondence:** Mateo Duque-Villegas (mateo.duque@mpimet.mpg.de)

**Abstract.** Over the last hundreds of millennia natural rhythms in Earth's astronomical motions triggered large-scale climate changes and led periodically to humid conditions in much of North Africa. Known as African humid periods (AHPs), such times sustained river networks, vegetation, wildlife and prehistoric settlements. Mechanisms, extent and timing of the changes still cannot be completely outlined. Although AHPs along glacial cycles are recognizable in long marine sediment records,

the related land cover changes are difficult to reconstruct due to scarcity of proxy data over the continent. Moreover, most available information covers only the latest AHP during the Holocene. Here we use a comprehensive Earth system model to look at additional, much earlier, possible cases of AHPs. We simulate the full last glacial cycle, aiming to reproduce the last four AHPs as seen in available proxies. The simulated AHPs seem in broad agreement with geological records, especially in terms of timing and relative strength. We focus on the simulated vegetation coverage in North Africa and we detect a dominant

change pattern that seems to scale linearly with known climate forcing variables. We use such scaling to approximate North African vegetation fractions over the last eight glacial cycles. Although the simple linear estimation is based on a single mode of vegetation variability (that explains about 70 % of the variance), it helps to discuss some broad-scale spatial features that had been only considered for the Holocene AHP. Extending the climate simulation several millennia into the future reveals that such pattern scaling breaks when greenhouse gases become a stronger climate change driver.

## 1   Introduction

Broad swings in environmental conditions over North Africa have happened for at least the last 11 million years (Crocker et al., 2022). What is now the hyper-arid Sahara was formerly at times a much wetter region, adequate for sustaining perennial water bodies and vast expanses of vegetation cover (Gasse, 2000; deMenocal et al., 2000). Such times, known also as African humid periods (AHPs), were recorded as quasi-periodic intervals of enhanced moist conditions in regional geological archives

(Grant et al., 2017, 2022). Establishment of humid climates came along with changes in the land surface that, via feedback loops, amplified the effects of changes in the Earth's orbit on the distribution of incoming solar radiation (Claussen et al.,



2017; Pausata et al., 2020). Extent and timing of the changes remain uncertain, owing to proxy data limitations and persistent discrepancies between existing geological records and climate simulations (Brierley et al., 2020; Yacoub et al., 2023).

The Holocene interglacial had the best known case of an AHP (deMenocal et al., 2000; Gasse, 2000), roughly between
11.7 ka to 4.2 ka (ka is thousands of years ago). Evidence collected for this period around North Africa revealed the changes in past lake highstands (Lézine et al., 2011), river networks (Drake et al., 2011; Blanchet et al., 2021), dust fluxes (McGee et al., 2013; Ehrmann et al., 2017), vegetation cover (Hély et al., 2014) and rainfall (Tierney et al., 2017b). In addition, numerical climate models were able to reproduce some of these changes during the Holocene AHP, and hinted at basic mechanisms underpinning AHP dynamics (Claussen et al., 2017). Nevertheless, what happened before the Holocene and how such mecha-
nisms played a role in deeper times is much more unknown. Repeated climate and land cover changes prevented widespread preservation of geological records older than the Holocene over land (Drake et al., 2018), and the costs of complexity and spatial and temporal resolutions limited in many cases the reach of simulations with comprehensive climate models.

What is known about pre-Holocene AHPs is mainly informed by signals recorded in marine sediments off the North African coastlines. Marine proxies, combined with the few surviving terrestrial signatures, and taking the Holocene AHP as a baseline,
make it possible to examine the cases of ancient AHPs along glacial cycles (Blome et al., 2012; Drake et al., 2013; Larrasoaña et al., 2013; Scerri et al., 2014). For instance, the multiple lines of geological evidence suggest that the previous interglacial during Marine Isotope Stage (MIS) 5e, near 127 ka, most likely had a much stronger AHP than the Holocene, in agreement with modelling results about the influence of changes in Earth's orbit. Likewise, marine proxies from around North Africa show other potential AHPs during the last few glacial cycles (e.g. Rossignol-Strick, 1983; Ehrmann and Schmiedl, 2021). However,
limitations in projecting marine proxy signals far inland, compounded with scant terrestrial sources, restricts our understanding about the amplitude of North African climate variability and AHP development. A clear outline of AHP changes is crucial to explain, for instance, regional geomorphology (Drake et al., 2022), ecological corridors during hominin evolution (Scerri et al., 2018) and long-term climate–land surface interactions (Pausata et al., 2020).

Climate models offer a theoretical window into possible spatial patterns of ancient AHPs. For instance, early experiments
with an atmosphere-only model already coarsely simulated the four AHPs since 150 ka (Prell and Kutzbach, 1987), as indicated by marine proxies. However, modelling long glacial cycles to include multiple AHPs requires a compromise between model complexity, simulation length and spatial resolution, which hinders the amount of detail that can be reproduced and evaluated. Studies that kept relatively complex models and higher spatial resolutions chose to either simulate discontinuous time slices (Singarayer and Burrough, 2015), or accelerate climate forcing rates (Kutzbach et al., 2008, 2020); in both cases reducing
the effects of high-frequency variability. Other studies opted for intermediate complexity models and large grid spacings to focus mostly on temporal patterns of average changes of AHPs (e.g. Tuenter et al., 2003; Tjallingii et al., 2008; Menviel et al., 2021). Although simulations broadly agreed with trends in the proxy records, underestimation of Sahara shrinking during AHPs and associated rainfall increments is a consistent problem across models, which prevents further understanding on the scale of the changes (Tierney et al., 2017b; Brierley et al., 2020). Simulating a full glacial cycle with interactive climate and



vegetation components at enough spatial resolution may reveal dominant AHP patterns underlying geological records or behind data–model mismatches.

We use a comprehensive climate model to perform a transient global climate simulation since 134 ka, at an affordable spatial resolution to see subcontinental changes in vegetation cover of North Africa. The main goal is to find general trends in simulated patterns during AHPs of the last glacial cycle. We evaluate different AHP responses and relate them explicitly with large-scale climate drivers: (1) Earth's orbital parameters, (2) radiative effects of levels of atmospheric greenhouse gases (GHG) and (3) high-latitude ice sheets. These drivers are known to set multiple controls on localized rain-bringing mechanisms (Dallmeyer et al., 2020; Blanchet et al., 2021), which could lead to diverse AHP biogeographies. Assuming local environments always reacted similarly to climate forcing, it is possible that AHP patterns could be approximated as a function of known forcing values. We explore such a connection between the drivers and the vegetation patterns they induce during AHPs. In addition, we also simulate and briefly assess potential future evolution of North African vegetation. In the following sections we describe the climate model, explain the experimental set-up, present results focused on North African vegetation cover during AHPs and discuss the possibility of forcing-based pattern scaling.

## 2 Methods

### 2.1 Model description

We use the Max Planck Institute for Meteorology Earth System Model (MPI-ESM; Giorgetta et al., 2013) version 1.2 (Mauritsen et al., 2019). It couples dynamical components representing the general circulations of the atmosphere (ECHAM6.3; Stevens et al., 2013) and ocean (MPIOM1.6; Jungclaus et al., 2013), as well as changes in the land surface and vegetation cover (JSBACH3.2; Reick et al., 2013). The standard release of this model version participated in the Coupled Model Intercomparison Project Phase Six (CMIP6; Eyring et al., 2016). Implementation of the coarse resolution set-up T31GR30 (about $3.75°$ horizontally) with 31 atmospheric and 40 oceanic levels enables long climate simulations at about maximum 700 simulated years per day (Mikolajewicz et al., 2018). Long simulations over several glacial cycles require updates during runtime of the land-sea mask, glacier mask and river routing, that align with reconstructions of ice sheet and sea level changes. This is done once every decade as described in Meccia and Mikolajewicz (2018) and Riddick et al. (2018). The specific model set-up is actively maintained and performs well within the range of current general circulation models (Kapsch et al., 2022).

How the model computes natural vegetation cover is particularly relevant for this study. A detailed description is given in Reick et al. (2013) and a performance assessment is in Brovkin et al. (2013). Briefly, the model uses a "mosaic" approach where each land surface grid cell is tiled to be occupied by fractions of predefined plant functional types (PFTs). PFTs include eight cover classes between woody (tropical and extra-tropical trees plus shrubs) and herbaceous (C3 and C4 grasses) vegetation, each with different turn-over timescales. When one vegetation type dies via natural causes (age) or via disturbances (fire or wind-throw), the "void" it leaves is filled (or "colonized") by growth of the most competitive PFT in terms of net primary productivity (NPP), which is a function of climate and atmospheric $CO_2$. Seeds for all PFTs are assumed to be universally





present. Fractions of PFTs can only occupy the portion of a grid cell that is hospitable for growth, which is determined based on having sufficiently productive (NPP) vegetation over consecutive years. The distribution of inhospitable fractions across the globe corresponds to hot and cold deserts. Vegetation cover in the model has immediate effects on the exchange processes between land surface and the atmosphere, which can subsequently alter climate patterns and feedback on vegetation (Brovkin et al., 2009).

## 2.2 Experiments

We simulate a transient Earth system response to prescribed forcing for the last $134 \, \text{kyr}$ (kyr is thousand years as in duration). Almost entirely as a single continuous run ($120 \, \text{ka}$ to $0 \, \text{ka}$), except for a 14-kyr interval spliced a posteriori to extend analysis until $134 \, \text{ka}$ (details about splicing in Appendix A). Earth's orbital parameters are computed from Berger (1978). Atmospheric levels of GHG ($CO_2$, $CH_4$ and $N_2O$) are taken from the ice core records compilation of Köhler et al. (2017), and are updated every 10 model years as decade means. Ice sheets from the GLAC-1D dataset (Tarasov and Peltier, 2002; Tarasov et al., 2012, 2014; Briggs et al., 2014; Abe-Ouchi et al., 2013) are interpolated spatially and temporally to suit model resolution and are prescribed also at decennial time steps, with corresponding adjustments in topography, bathymetry and river routing (Meccia and Mikolajewicz, 2018; Riddick et al., 2018). Details about the GLAC-1D dataset extension until before the last interglacial are given in Menviel et al. (2019).

Figure 1 (left) compiles a set of variables directly linked to forcing factors: orbitally driven changes in low-latitude insolation in terms of the so-called orbital monsoon index (Fig. 1a); the Northern Hemisphere total volume of ice sheets (Fig. 1b); and the GHG radiative forcing change relative to the pre-industrial era (PI, 1850 Common Era or CE; Fig. 1c). The monsoon index of Rossignol-Strick (1983) is an insolation difference between the Northern Tropic ($23.4° \, \text{N}$) and the equator during the caloric summer season defined by Milankovitch (1941). Changes in the radiative forcing of GHG are based on an "equivalent-$CO_2$ concentration" explained in Ganopolski et al. (2010). We ignore any land use and atmospheric aerosols remain at pre-industrial values (Kinne et al., 2013).

Additionally, we run two transient experiments $10 \, \text{kyr}$ into the future. In both, the ice sheets remain unchanged at present-day conditions, while the orbital forcing continues to change according to Berger (1978). For GHG we choose two alternative Shared Socioeconomic Pathway (SSP) scenarios from the Sixth Assessment Report (AR6) of the Intergovernmental Panel on Climate Change (IPCC): SSP1-1.9 and SSP3-7.0 (IPCC, 2021). The two scenarios are selected simply as contrasting cases of potential future GHG dominance. The GHG data for these scenarios come from Meinshausen et al. (2020) until the year $2500 \, \text{CE}$, and onwards from previous modelling output of Brovkin et al. (2012), similarly as in Kleinen et al. (2021).







**Figure 1.** Transient simulation of the last glacial cycle. Forcing variables (left) represent prescribed conditions driving the experiment: (a) orbital parameters following Berger (1978) are shown using the orbital monsoon index of Rossignol-Strick (1983); (b) ice sheets from the GLAC-1D dataset (references in main text) are shown with the total ice volume in the Northern Hemisphere (NH); and (c) greenhouse gases (GHG) from Köhler et al. (2017) are shown as radiative forcing (RF) changes from pre-industrial era (PI). Simulation results compared with proxy records (right): (d) changes in global mean near-surface temperature relative to PI, compared to isotopic data from Lisiecki and Raymo (2005); (e) meridional overturning circulation (MOC) in the Atlantic Ocean (1000-m depth, basin mean at $26°$ N); and (f) changes in precipitation in the Tropical North Atlantic (TNA; defined in main text) compared to marine sediments reflectance ($L^\star$; smoothed with a third-order low-pass Butterworth filter with a cutoff frequency of $1/5\,\mathrm{kyr}^{-1}$) data from Deplazes et al. (2013), both related to shifts of the Inter-Tropical Convergence Zone (ITCZ). We include labels for marine isotope stages (MIS) according to Railsback et al. (2015).

## 3 Results

### 3.1 Earth system response

Figure 1 (right) shows some globally relevant mean climate variables. Global climate is the warmest during the MIS 5e interglacial near $127\,\mathrm{ka}$ (Fig. 1d). About $0.7\,\mathrm{K}$ warmer than simulated pre-industrial temperature, the last interglacial warming





matches proxies in the sign of change, but the amount could be underestimated by up to $1.3\,\mathrm{K}$ (Turney et al., 2020), similarly
as in other climate models (Lunt et al., 2013; Otto-Bliesner et al., 2021). Minimum average temperature occurs during the
Last Glacial Maximum (LGM) around $24\,\mathrm{ka}$, with roughly $4.4\,\mathrm{K}$ colder than during PI, which is not far from recent estimates
ranging from $5\,\mathrm{K}$ to $7\,\mathrm{K}$ (Tierney et al., 2020; Osman et al., 2021). Amplitude and timing of temperature changes seem in
agreement with the isotopic data from Lisiecki and Raymo (2005).

Ocean dynamics maintains a meridional overturning circulation (MOC) in the Atlantic Ocean around $19\,\mathrm{Sv}$ (Fig. 1e), in-
cluding millennial-scale fluctuations related to freshwater input from glacier run-off as the ice sheets change. Strongest MOC
variability occurs during late deglaciation phases around $130\,\mathrm{ka}$ to $128\,\mathrm{ka}$ and $15\,\mathrm{ka}$ to $9\,\mathrm{ka}$. The last deglaciation includes an
especially strong meltwater pulse near $14.4\,\mathrm{ka}$ that nearly collapses the Atlantic MOC in the simulation (Kleinen et al., 2023).
After this meltwater pulse, the remainder of the simulation includes some unexpected Atlantic MOC variability during the early
Holocene. The amplitude of this early-Holocene MOC variability is not represented in the meltwater release from the ice sheet
reconstruction, and it was not seen in previous deglaciation experiments with the same model (e.g. Kapsch et al., 2022; Kleinen
et al., 2023). In our simulation it appears to be related to a slight difference in the initial condition of ocean volume that results
in some shallow regions in the Arctic Ocean. The Atlantic MOC changes are associated with drops in global temperature and
in precipitation in the tropics (see Fig. 1d and f).

As a proxy for large-scale circulation changes in the atmosphere, we also show in Fig. 1f the simulated precipitation changes
in the Tropical North Atlantic (TNA; $10°\,\mathrm{N}$ to $20°\,\mathrm{N}$, $80°\,\mathrm{W}$ to $20°\,\mathrm{W}$), where changes are coupled to the movement of the
Inter-Tropical Convergence Zone (ITCZ). Unlike global temperature, these changes follow more closely the oscillations of the
orbital monsoon index, and the timing of the minima of precipitation (when the ITCZ is southernmost) broadly agrees with the
Cariaco Basin record of Deplazes et al. (2013). However, the simulation is slightly offset and the proxy record shows much
more variability, probably related to regional or local scale signals in the data. Besides model precipitation is also smoother
after the regional averaging.

## 3.2 African humid periods

### 3.2.1 Timing and strength

Marine sediments off the coast of North Africa reveal four AHPs in the last $134\,\mathrm{kyr}$ (Fig. 2a–d). AHPs appear as times of
pronounced depletion of deuterium ratios in East Africa (Tierney et al., 2017a), because increased rainfall dilutes isotopic signal
(Fig. 2a). Also as times of reduced Saharan dust output to the Atlantic Ocean (Skonieczny et al., 2019) and Mediterranean Sea
(Ehrmann and Schmiedl, 2021), because increased soil-stabilising vegetation cover reduces dust deflation (Fig. 2b, c). Likewise
during AHPs, increased river discharge into the Mediterranean stratified seawater and promoted sapropel bands formation (Fig.
2d). In spite of notable differences in location and methods amongst proxies, AHPs stand out rather consistently.

Focusing on the kaolinite/chlorite proxy (Fig. 2c), it reveals details about the intensity of the AHPs, according to the am-
plitude of dust pulses, which depends on the degree of (rain-fed) weathering that accumulates the kaolinite mineral in water

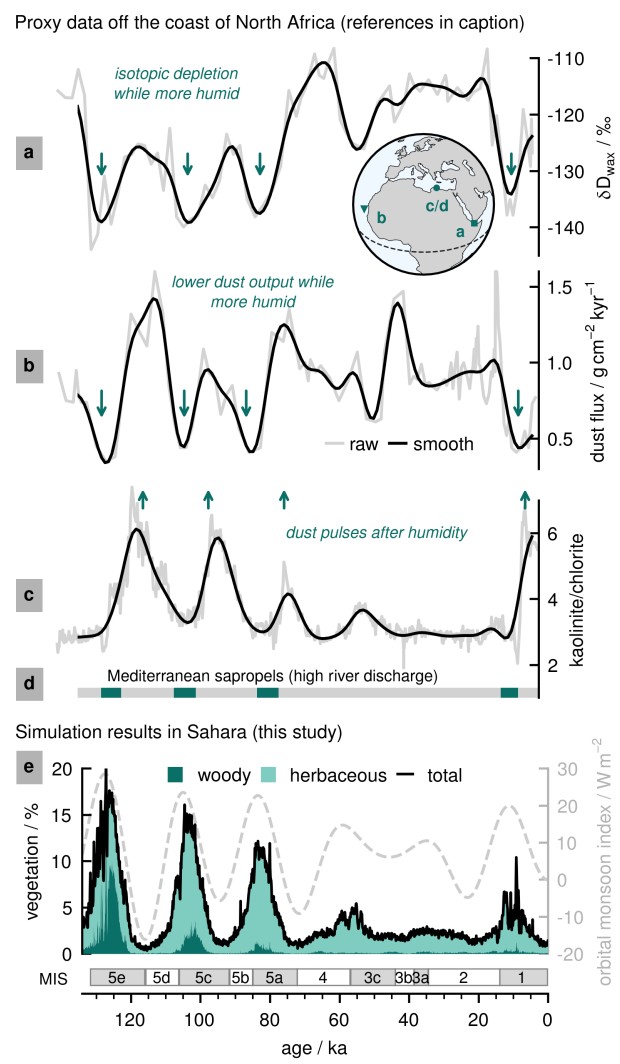

**Figure 2.** North African climate in the last glacial cycle as seen in (a–d) marine proxies offshore and (e) average simulated vegetation coverage (woody, herbaceous and total) in the Sahara (defined in main text). Proxy data (grey) are interpolated evenly at $1\,\mathrm{kyr}$ resolution and smoothed (black) with a third-order low-pass Butterworth filter with a cutoff frequency of $1/10\,\mathrm{kyr}^{-1}$: (a) deuterium ratio changes ($\delta$D) in terrestrial plant lipids (wax) in core RC09-166 near East Africa (Tierney et al., 2017a); (b) $^{230}$Th-normalized Saharan dust flux into the Atlantic Ocean according to core MD03-2705 near West Africa (Skonieczny et al., 2019); (c) influx of fine-sized dust to the Eastern Mediterranean Sea according to the ratio of kaolinite and chlorite minerals in core SL99 and (d) its sapropel sequence (Ehrmann and Schmiedl, 2021). In (e) we also show a secondary y-axis with the orbital monsoon index. Proxy sites are shown in an inset globe.

bodies while an AHP is active (Ehrmann et al., 2017). Accordingly, in Fig. 2c, the AHP of MIS 5e stands out clearly as the strongest (largest dust pulse near $118\,\mathrm{ka}$), while the estimated magnitude of other AHPs based on kaolinite/chlorite has been indicated to vary with the core-to-shore distance (Ehrmann and Schmiedl, 2021).





Figure 2e shows simulation results of total vegetation fraction in the Sahara ($20°$ N to $30°$ N, $20°$ W to $25°$ E). The simulation
appears well in phase with trends in the geological records. We find four AHP-like changes in North African climate, seen as
large peaks in average Saharan vegetation. The vegetation peaks track the orbital monsoon index but with a small lag of about
$1\,\mathrm{kyr}$. The lag probably appears because the orbital monsoon index is tied to the summer season only and to the Northern Tropic
($23.4°$ N), while the simulated vegetation response is also affected by what happens in the extra-tropics, not only in the summer
but also during the winter season. Saharan vegetation peaks happen sometime between peak boreal summer temperatures and
mild boreal winter temperatures.

Simulated AHPs appear also as times of herbaceous cover expansion and woody types invasion of the Sahara. The AHP of
MIS 5e is the strongest event with above $15\,\%$ Saharan vegetation cover, followed by two weaker AHPs during MIS 5c and 5a,
both clearly over $10\,\%$ total vegetation fraction. In the simulation the Holocene has the weakest AHP, barely approaching $10\,\%$
of vegetation in the Sahara. Such a weak Holocene AHP is unexpected according to previous applications of the same model
to the last deglaciation (e.g. Dallmeyer et al., 2022).

In our simulation the Holocene climate is largely affected by pronounced Atlantic MOC variability (cf. Fig. 1e), whose
amplitude is greater than prescribed with the ice sheet reconstruction changes. The large Atlantic MOC fluctuations appear
connected with a small ocean volume difference in the initial condition of the experiment. To diagnose the weak Holocene
AHP we include Appendix B, where we show the global climate anomalies with respect to pre-industrial temperature and
precipitation. The effects of the Atlantic MOC variability are clearly seen in the early Holocene having a much colder North
Atlantic region (Fig. B1f). This keeps the ITCZ and related rainbelt southwards and explains the anomalously weak Holocene
AHP (see Tjallingii et al., 2008).

Nevertheless, strength of AHPs in Fig. 2e is meant only relative to each other, since an overall Sahara greening below $20\,\%$ is
still substantially less than previous estimates from intermediate complexity models (e.g. Menviel et al., 2021; Duque-Villegas
et al., 2022), as well as less than estimates from the same model but implemented at higher spatial resolution (Dallmeyer et al.,
2020). In Appendix C there is a brief comparison of the simulated climate near present-day (mean values for 1990–2020 CE)
with current observational products. We find the model simulates less vegetation in the Sahel (despite neglecting land use in the
simulation) and is overall in North Africa much drier (about $1\,\mathrm{mm\,d^{-1}}$) than inferred from the observational data. Explaining
the mismatch between different complexity models or why there is a dry bias in the model at this spatial resolution is beyond
the scope of this work, thus we focus on relative changes rather than absolute values.

### 3.2.2 Vegetation patterns

The spatial extent of an AHP also tells its intensity. Figure 3 shows maximum vegetation cover fraction in North Africa during
AHPs and changes from PI. In this case AHP values are selected from time slices around the vegetation optimum in the Sahara
(see guide in figure). It is important to note that it is not trivial to define AHPs for the whole North Africa, since it is a large
region that responds with spatio-temporal heterogeneity (see Dallmeyer et al., 2020). For this reason we show maximum values
that may happen at different times (a few centuries earlier or later) for different sites (grid points) within an AHP time slice.



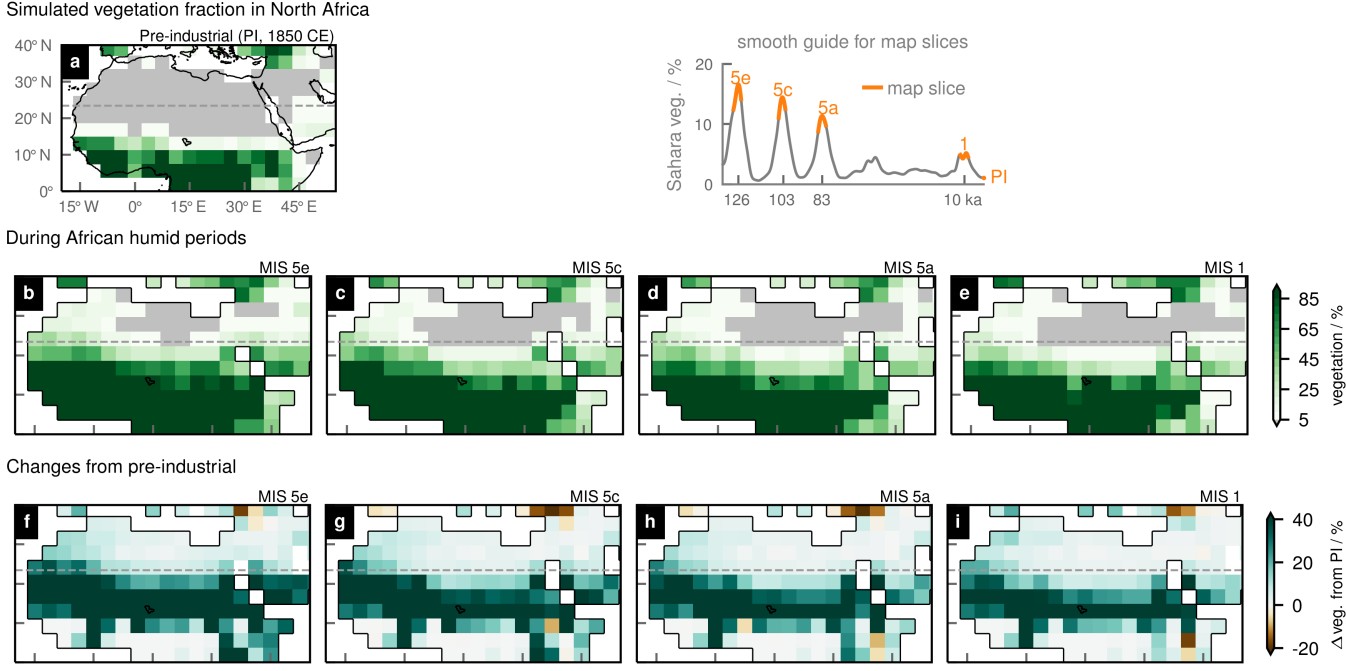

**Figure 3.** North African vegetation coverage. (a–e) Maximum vegetation coverage during pre-industrial (PI) era and AHPs, as well as (f–i) the relative changes from PI. A time series inset (top right) has marine isotope stages (MIS) labels to highlight mapped AHP time slices spanning about 4 kyr. In maps a horizontal line shows the Northern Tropic (23.4° N). Only the PI map shows modern coastlines, while others show the coastlines according to reconstructed sea-level changes. Lake Chad is shown in all maps as a point of reference.

Related to this caveat there is an interesting distinction about the changes in Fig. 3 between areas that are already "green" for pre-industrial conditions (like below the Sahel), which become "greener" during AHPs, and areas that are barren at PI which *turn* "green" during AHPs (like inside the Sahara).

Pre-industrial conditions in the simulation (Fig. 3a) already seem to underestimate zonal patterns of vegetation (like present-day dry bias in Appendix C), resulting in little vegetation in the Sahel (near 15° N) and an exaggerated Dahomey Gap (dividing the Coast of Guinea) that is much less conspicuous in higher resolution versions of the same climate model (e.g. Bathiany et al., 2014). During AHPs there is a continent-wide increment in vegetation, closely approaching the Northern Tropic (23.4° N) and effectively shrinking the Sahara (Fig. 3b–e). The largest changes occur around the Sahel, within 10° N to 20° N, with over 40 %
increments (Fig. 3f–i). The smallest changes seem to be cornered in northeastern Sahara. Near the equator, in places where there is already close to 100 % vegetation cover under pre-industrial forcing, there are also small or no changes during AHPs.

AHP patterns in Fig. 3b–e seem to become weaker sequentially, starting with more intense changes during the AHP of MIS 5e on the left, progressing towards milder ones in the Holocene (MIS 1) on the right, and with two intermediate cases in between. Most apparent is the fading of intense changes around the Sahel, as well as a less pronounced northward push along



the west coast. Such step-wise (ignoring long time gaps between AHPs) weakening of AHPs is also evident in the average vegetation fraction in the Sahara (see Fig. 2e). Incremental change occurs in the same direction as the orbital forcing (see Fig. 1a), therefore suggesting the possibility of a linear scaling of a baseline pattern with climate forcing variables. In the following section we explore such linear relationship for vegetation cover fractions.

## 3.3   Spatio-temporal analysis

AHPs seem quasi-periodic in response to natural rhythms of the Earth system. Possibly there are preferred modes (patterns) of AHP variability (i.e. climate and biogeographies) that regularly appeared in the past during AHP development. In such case it may be assumed that the broadest patterns would tend to scale proportionally to large-scale climate forcing conditions. We investigate this possibility of pattern scaling in the simulated vegetation fractions in North Africa during the last glacial cycle, with the main focus on the potential to capture with this method the simulated AHP patterns of Fig. 3.

To find a latent spatial pattern underlying North African vegetation variability (including AHPs as peak amplitudes) we perform an empirical orthogonal function (EOF) analysis (also known as principal components analysis). We construct a covariance matrix with the changes (from PI) of annual maximum vegetation cover fraction for grid cells inside North Africa ($p = 126$). Values are century means (for the time dimension $n = 1340$). The matrix is centred about its mean (remove trend in changes from PI) and decomposed with a singular value decomposition to obtain its eigenvectors ordered according to

explained variance. After decomposition we only use the first eigenvector ($EOF_1$) which, together with its corresponding time series ($PC_1$), explains the greatest variability.

Figure 4 shows the results of the EOF analysis. The dominant pattern of variability ($EOF_1$) explains about 73 % of the variance in vegetation fraction changes (Fig. 4a). It is a mostly zonally coherent pattern with an action centre largely inside $10° N$ to $20° N$. Near eastern Africa the pattern adopts a more meridional structure, between $30° E$ to $45° E$. According to

its corresponding $PC_1$ (smoothed in Fig. 4b) this mode is related to uniform changes in vegetation fraction with respect to pre-industrial conditions, which are positive and of great amplitude usually during warm interstadials (shaded MIS boxes). $PC_1$ also seems in phase with the orbital forcing monsoon index (compare Fig. 1a). The phasing with the monsoon index is a good indicator of the possibility of estimating $PC_1$ as a function of external climate forcing.

Figure 4b also shows a linear model that combines the orbital monsoon index (lagged $1\,kyr$ and symbolized as $O_{lag1k}$), the

GHG radiative forcing (symbolized as G), and the Northern Hemisphere ice sheet volume (symbolized as $I_{NH}$). The model is fit to the smooth $PC_1$ using an ordinary least-squares method. We choose to lag the orbital monsoon index as a predictor because of the lag in the vegetation peaks explained in Section 3.2.1. Fitting is done several times in different parts of the time series, using a sliding (every $10\,kyr$) window of $63\,kyr$ (about three precession cycles), which explains the uncertainty in the regression parameters.

Overall, the linear model ($_{forcing-based}PC_1$ in Fig. 4b) estimates a similar trend to the smooth $PC_1$, with some uncertainty in the parameters that leads to some spread. It is possible that other more complicated fits could reproduce the smooth $PC_1$ pattern



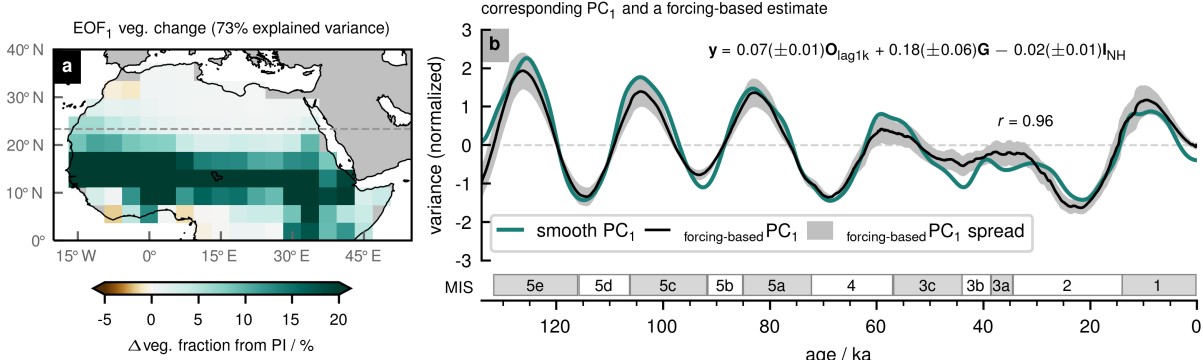

**Figure 4.** EOF analysis of North African changes in vegetation fraction from pre-industrial (PI) era: (a) leading variability mode ($EOF_1$) and (b) corresponding $PC_1$ time series scaled to unit variance. $PC_1$ is smoothed with a third-order low-pass Butterworth filter with a cutoff frequency of $1/5\,\mathrm{kyr}^{-1}$. (b) includes also a forcing-based estimate for the smooth $PC_1$ (central trend and spread due to uncertainty of regression parameters in parentheses) whose linear regression model equation is shown, where: $O_{lag1k}$ is orbital forcing monsoon index lagged 1 kyr (cf. lag in Fig. 2e), **G** stands for GHG radiative forcing change from PI, and $I_{NH}$ is ice sheet volume in the Northern Hemisphere. Modern coastlines are shown in (a).

better, or with less spread, but we restrict our analysis to the simplest fit. In the linear equation it is interesting to note that for the last glacial cycle the standard deviation of $O_{lag1k}$ is close to $10.1\,\mathrm{W\,m}^{-2}$, while that of G is around $0.8\,\mathrm{W\,m}^{-2}$, and that of $I_{NH}$ around $11.0\,\mathrm{1e6\,km}^3$. Multiplying these values by the absolute value of their respective fit parameters (0.07, 0.18 and 0.02), it shows that orbital, GHG and ice-sheet variables influence about $66\,\%$, $13\,\%$, and $21\,\%$, respectively, of the outcome of the linear model. The dominance of the orbital forcing is likewise clearly visbile in the oscillating shape of the linear fit. The fitting seems to perform most poorly after $60\,\mathrm{ka}$, when glacial conditions become more important as they approach their maximum.

## 3.4 Pattern scaling

We can use the results of the previous section to obtain forcing-based estimates of the vegetation changes (relative to PI) in North Africa. Our method resembles a pattern scaling, where an invariant spatial pattern is assumed to scale linearly with a known forcing factor. The relationship to the climate forcing is based on the curve fit to the smooth $PC_1$ time series in Fig. 4b, using the linear sum of the orbital monsoon index (lagged 1 kyr), GHG radiative forcing and Northern Hemisphere ice sheet volume. Because of the use of the principal component time series, another name for this method is also principal component emulator (Wilkinson, 2010), which has been used before also in palaeo-climate modelling contexts (e.g. Holden et al., 2019). Using $EOF_1$ and $_{forcing-based}PC_1$ we can approximate mean vegetation changes having forcing variables as predictors. We are especially interested in estimating the mean changes during AHPs.





In Fig. 5 we evaluate the skill of the pattern scaling to predict the simulated vegetation changes (from PI) during the same AHP time slices of Fig. 3. This means the climate model output is acting as reference and errors come from scaled minus simulated fields. We find that for the AHPs of MIS 5 (Fig. 5a, b, and c) the pattern scaling underestimates the vegetation changes inside the Sahara, which is most noticeable in western Africa. Only for the Holocene (Fig. 5d) there is a slight overestimation of vegetation growth during the AHP. That our approach projects only limited changes in the Sahara (e.g. north of $20°$ N) is expected because the $EOF_1$ pattern (Fig. 4b) has more pronounced loadings in the Sahel (near $15°$ N). We found other EOF patterns (not shown) that could have provided more changes in the Sahara, however, their associated temporal patterns were not simple to link to the external climate forcing. Nonetheless, considering the magnitude of some of the changes in North Africa whole (above $+40\,\%$ in Fig. 3), root-mean-square errors (RMSE) below $\pm\,10\,\%$ vegetation cover change in Fig. 4a–d may be seen as acceptable.

Furthermore, Fig. 5e shows the average performance of the pattern scaling. We find the similar outcome that the method predicts intermediate values between the rather-strong AHPs during MIS 5 and the weaker Holocene AHP. We hypothesize that if there had not been the unexpectedly strong Atlantic MOC variability during the early Holocene (also seen in Fig. 5e), which weakened the Holocene AHP, our method would find a curve fit able to predict slightly larger values, closer to those during MIS 5. It is also interesting to see that the scaling spread, due to uncertain regression parameters, keeps the climate simulation largely within the reach of the pattern scaling.

## 3.5 Late Quaternary AHPs

The pattern scaling method opens an interesting possibility to quickly obtain a rough estimate of the vegetation fraction distribution in much of North Africa for other times for which reliable forcing data are available (and yet may lie outside of current modelling capabilities). It seems best suited for the Sahelian vegetation response (where the main loading of the EOF pattern lies), however, it still provides part of the vegetation change everywhere in North Africa (e.g. the Sahara), like a minimum probable estimate of the change. Two main assumptions underlying this approach are that (1) the $EOF_1$ pattern captures much of the large-scale pattern of regional variability, and (2) the forcing-based linear model effectively emulates the long-term time-dependence of variation at that scale and the climate forcing. Although it may likely be biased towards a single glacial cycle, proxy data suggests the last glacial cycle could be representative of large-scale climate changes happening for at least the last eight glacial cycles (Lüthi et al., 2008). Therefore, it is plausible that the pattern scaling generalizes to multiple other cycles.

We use the scaling approach to estimate changes in North Africa since $800\,\mathrm{ka}$ using available forcing data. The orbital monsoon index is computed for this interval based on the parameters of Berger (1978). The GHG radiative forcing and ice sheets data come from Ganopolski and Calov (2011), who compiled gases data from Antarctic ice core records (Petit et al., 1999; EPICA Community Members, 2004), and simulated the ice sheets with a coupled intermediate complexity model and ice-sheet model. Figure 6a–c shows this longer set of forcing variables, where the new GHG and ice sheets data agree with the forcing used in our simulation and to fit the linear model for the overlapping period of the last $134\,\mathrm{kyr}$. In Fig. 6a we also





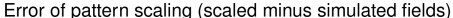

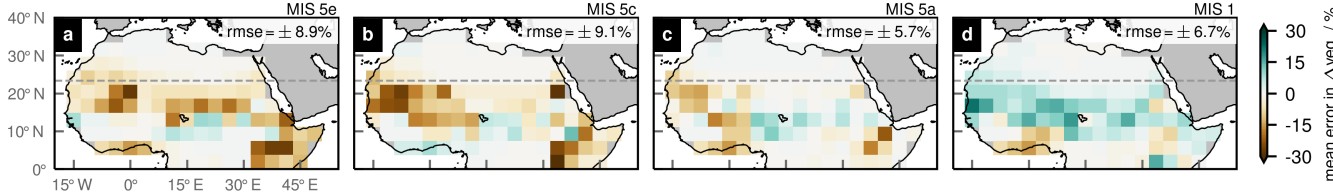

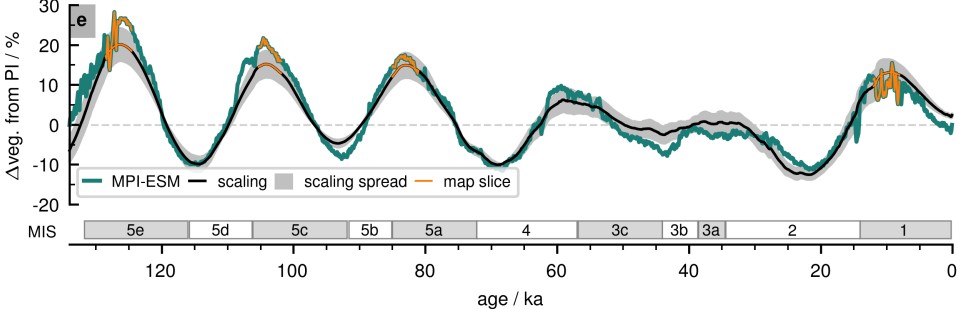

**Figure 5.** Assessment of pattern scaling method for estimation of change in vegetation fraction from pre-industrial (PI) era: (a–d) mean spatial errors during the AHP time slices highlighted in (e); and (e) domain-wide average performance. This assessment uses the MPI-ESM simulated output as a reference (i.e. errors are scaled minus simulated patterns). In (a–d) the root-mean-square errors (RMSE) are shown. Maps include modern coastlines only as a spatial reference.

show, as a guide, the orbital monsoon index threshold at $19.8\,\mathrm{W\,m^{-2}}$, which relates to occurrence of AHPs as detected by Rossignol-Strick (1983) and discussed more recently in Duque-Villegas et al. (2022).

Evidence of the AHPs of the last $800\,\mathrm{kyr}$ is available in marine proxies. Figure 6d and e depict Mediterranean proxy records linked to past AHPs: the sapropel record of Emeis et al. (2000) and the bulk composition elements in the sediments core of Grant et al. (2022), respectively. Both long records agree for many events, although the geochemical analysis of Grant et al. (2022) detects several more AHPs. We compare such records with our simulation and pattern scaling approach.

Figure 6f shows the last eight glacial cycles of average vegetation changes in North Africa according to the pattern scaling, according to the forcing variables shown in Fig. 6a–c. Figure 6f, also includes a reference line set at $14\,\%$ of average vegetation change, based on the AHPs of the last glacial cycle, which is useful to guess potential more ancient AHPs. According to this reference guide two other potential AHPs stand out, possibly similarly as strong as the AHP during MIS 5e: one happening around $580\,\mathrm{ka}$, within MIS 15a, and another one near $330\,\mathrm{ka}$, within MIS 9e. The Mediterranean records also clearly contain these two events (Fig. 6d and e).

The average vegetation changes in Fig. 6f also compare well with the BIOME4 dataset generated with the statistical method in Krapp et al. (2021), although the reconstructed spatial patterns have differences in the zonal extent of the changes (not shown). Moreover, Figure 6g and h show the estimated changes from PI and vegetation fraction pattern, respectively, during





**Figure 6.** North African vegetation fraction changes over the last $800\,\mathrm{kyr}$. (a–c) set of forcing factors used for scaling, including in (a) the Rossignol-Strick (1983) threshold for sapropel formation at $19.8\,\mathrm{W\,m^{-2}}$; (d) composite sapropel record (cores ODP160-966/7), with wide dark bands for clear sapropels and narrow light bands for oxidized or missing sapropels (Emeis et al., 2000). (e) inferred AHPs (colour bands) from bulk elements in sediments core ODP160-967 (Grant et al., 2022). (f) pattern scaling results for area average changes from pre-industrial (PI) era. A horizontal line at $14\,\%$ is an empirical threshold for counting potential AHPs based on AHPs of the last glacial cycle. (g, h) scaled patterns during potential ancient AHP of MIS 15a, showing modern coastlines even though they may differ at that time.

one of the potentially strongest AHPs (near $580\,\mathrm{ka}$). It is only one example of the pattern scaling for beyond-simulated AHPs. When considering such an ancient AHP caution is important and it is useful to know, for instance, that the land-sea mask and topography may differ from the one used to calculate the $\mathrm{EOF_1}$ pattern.



Based on the empirical threshold line in Fig. 6f, the scaling approach can identify most clearly some 19 (potential and
known) AHPs. This number agrees with the 21 possible AHPs based only on the orbital monsoon index threshold (see Fig.
6a) detected by Rossignol-Strick (1983). Considering the spread in the parameters of the linear fit, the number could be as
low as eight (including the Holocene AHP despite its minimum spread value being under the empirical threshold) or as high
as 21 (like the monsoon index threshold would predict). Most of the candidate AHPs would be in agreement with signals in
Mediterranean sediment cores (Fig. 6d and e). A similar amount of 20 AHPs was reported from many independent equilibrium
simulations along this time range (Armstrong et al., 2023). Likewise, a data synthesis by (Larrasoaña et al., 2013) indicated 22
potential AHPs for this interval. The refined geochemical analyses by Grant et al. (2022) which read subtle humidity variations
estimated about 30 episodes of Saharan greening since $800\,\mathrm{ka}$ (see Fig. 6a). However, some of the events could perhaps be
grouped into single cases of AHPs.

### 3.6 Future AHPs

Previous studies pointed out the differences between orbitally dominated AHPs in the past, and predicted, GHG-modulated,
AHPs in upcoming millennia (Claussen et al., 2003; D'Agostino et al., 2019; Duque-Villegas et al., 2022). For the next $100\,\mathrm{kyr}$
Earth keeps a low eccentricity orbit (a cycle of about 400-kyr) that hinders the possibility of a new AHP inception. At most there
could be some mild greening in North Africa near $66\,\mathrm{kyr}$ into the future (Duque-Villegas et al., 2022). The low eccentricity
reflects in the small variation in the orbital monsoon index in Fig. 7a, which stays far from the threshold of about $19.8\,\mathrm{W\,m^{-2}}$
for the onset of an AHP detected by Rossignol-Strick (1983) and Duque-Villegas et al. (2022) (compare Fig. 7a with Fig.
6a). However, when compounded with increasing emission rates of GHG, the strengthening of the planetary radiative forcing
(as shown for scenarios in Fig. 7b) enhances humidity levels globally, possibly enabling an earlier greening in North Africa
(Duque-Villegas et al., 2022).

Scenarios in Fig. 7b have peak emissions in the near future (a few centuries) and later maintain a constant positive forcing,
leading to an average global warming of about $7\,\mathrm{K}$ (SSP3) and $1\,\mathrm{K}$ (SSP1). Figure 7c shows simulated North African vegetation
response to those scenarios. SSP3 seems to arrive at AHP conditions with a similar level as during MIS 5e (about $30\,\%$ average
change from PI), while SSP1 at about $10\,\%$ average vegetation change (which is smaller than during the Holocene) does not
resemble an AHP (note the threshold at $14\,\%$ in the previous section).

Figure 7c also shows the average performance of the pattern scaling for future times. It performs relatively well for the SSP1
scenario, but it greatly underestimates the SSP3 changes (even considering the parameter spread). This means the method is
rather conservative in terms of the GHG forcing, which could be related to the fitting being done within the narrow range of
GHG variability of the last $134\,\mathrm{kyr}$, or because the model includes also the glacial changes in the ice sheets, which for these
future scenarios are assumed to remain invariant at present-day conditions. Moreover, our estimate method is bound to the
$\mathrm{EOF}_1$ pattern, which means it is not suited to account for greening related to possible future shifts in vegetation boundaries.

Figure 7d–g shows the patterns of vegetation change (from PI) that were dynamically simulated (with MPI-ESM), as well
as those estimated with the pattern scaling, near the orbital monsoon index local maximum at $9\,\mathrm{kyr}$ after present. This local

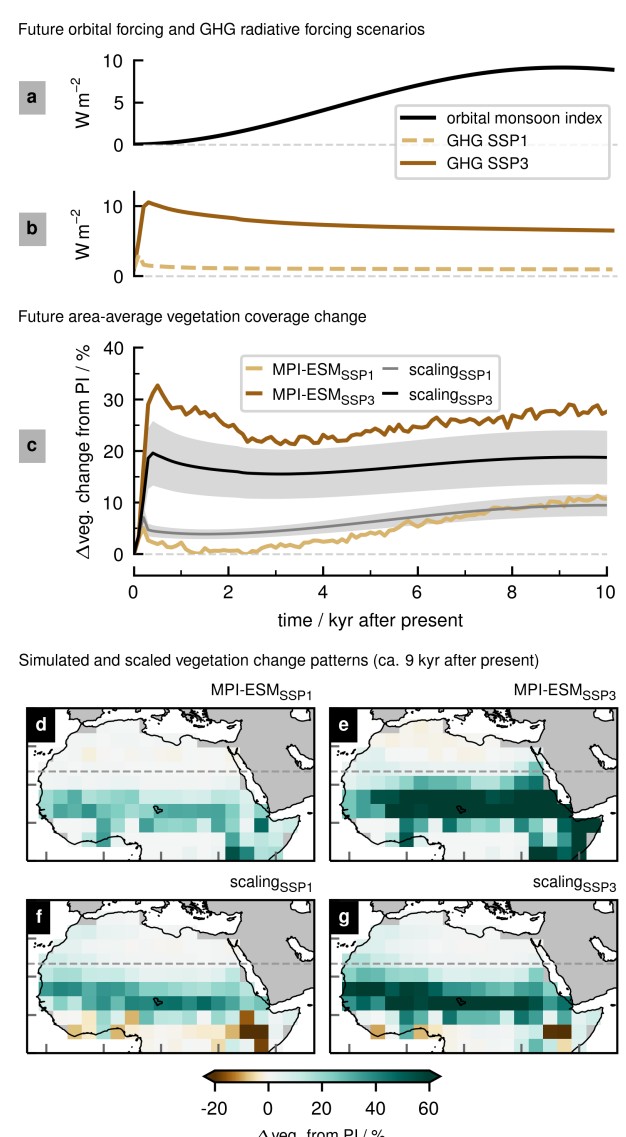

**Figure 7.** Future vegetation coverage change in North Africa: (a) orbital forcing monsoon index and (b) GHG radiative forcing scenarios; (c) simulated and scaling-based average vegetation fraction change from pre-industrial (PI) era, where grey bands show the scaling spread; and (d, e) mean simulated and (f, g) scaling-based vegetation fraction under forcing scenarios near 9 kyr after PI.

maximum in tropical insolation could see substantial greening in North Africa, potentially developing the next AHP, depending on the GHG forcing. Under the relatively low GHG forcing scenario SSP1, the pattern scaling (Fig. 7f) performs well in comparison to MPI-ESM (Fig. 7d). However, Fig. 7e shows that under the high GHG forcing scenario SSP3, the simulated

vegetation change pattern does not resemble any of its AHP predecessors, since it is missing the northward vegetation push along the west coast. Such difference agrees with previous comparisons of climate change projections and the Holocene AHP





(e.g. D'Agostino et al., 2019; Pausata et al., 2020). Therefore, our single-EOF pattern scaling approach, based only on late Quaternary vegetation changes, lacks skill to predict the vegetation patterns under high GHG forcing.

## 4    Discussion

We present results of a coupled atmosphere-ocean-vegetation simulation for the last glacial cycle. The simulation may be unique regarding model complexity, length and the absence of any acceleration technique. Other studies with similar times of interest opted for atmosphere-only models (Prell and Kutzbach, 1987), lesser complexity models (Blanchet et al., 2021; Menviel et al., 2021; Duque-Villegas et al., 2022), ensembles of discontinuous equilibrium simulations (Singarayer and Burrough, 2015; Armstrong et al., 2023), or acceleration methods (Kutzbach et al., 2008, 2020). In this case the simulation includes a realistic

representation of interactions across a broad spectrum of variability frequencies, which means disturbances can propagate across the globe to superimpose climate forcing and local effects. However, for the same reason it is not possible to implement a highly resolved grid in this long palaeo-experiment, which would enable a more refined comparison with the geological record and reconstructions, or it would be able to capture more clearly the shifts of vegetation boundaries (e.g. Dallmeyer et al., 2020).

We focus our analysis on vegetation cover in North Africa, where continental-scale climate changes produce four AHPs and a northward greening reaching into the Sahara. Simulated humidity changes align well with available terrestrial and marine proxies (Blome et al., 2012; Drake et al., 2013; Tierney et al., 2017a; Skonieczny et al., 2019; Ehrmann and Schmiedl, 2021), at least in terms of trends in deviations from pre-industrial conditions. In absolute terms, we find the climate model underestimates the extent of vegetation already at pre-industrial and present-day times (there seems to be a dry bias in this model at the

resolution we use), which for AHPs it means vegetation advancement into the Sahara is likely underestimated. Most of the simulated greening happens within the Sahel region. Especially the Holocene AHP appears largely underestimated in our simulation, because of the unexpectedly large Atlantic MOC fluctuations, which resulted in a much colder North Atlantic and Northern Hemisphere while this AHP developed.

Trends in the vegetation changes of the four simulated AHPs seem consistent with estimated features from the Holocene
humid event (Brierley et al., 2020; Dallmeyer et al., 2020). There is a northward progression of mostly zonal vegetation bands that extend east until about $40°$ E, with a slight northwest–southeast tilt related to further inland penetration of the West African monsoon system. Such tilted AHP development pattern is also detected in the simulations and reconstructions of Armstrong et al. (2023). Moreover, it was shown that palaeo-conditioning or re-tuning a climate model to improve palaeo-modelling results also led to a similar vegetation pattern during the Holocene (Hopcroft and Valdes, 2021). In our case, however, such tuning is

not applied. Yet because of a dry bias in the model at the implemented resolution, it is likely that the west-east differences in our vegetation change patterns are underestimated. Specifically during interglacials (MIS 5e and Holocene), simulated AHP greening appears much weaker than in the reconstructions of Larrasoaña (2021), and our results, in line with Armstrong et al. (2023), indicate the possibility of some desert cover remaining in a northeastern corner of the domain, thereby breaking the





zonal pattern of vegetation towards the north. Vegetation changes during MIS 5e align well with the precipitation changes in
the model–data comparison of Scussolini et al. (2019).

An interesting comparison of our pattern scaling can be made with the global biomes dataset generated by Krapp et al. (2021)
for the last $800 \, \text{kyr}$. In Krapp et al. (2021), authors also use model output for the last glacial cycle to create a linear regression
model that can extend simulated variables much further back in time until $800 \, \text{ka}$. Similarly, the predictor variables are forcing
terms like orbital parameters and atmospheric $CO_2$. However, in that case the approach was implemented globally and the
estimated changes in North Africa seem much more zonal in comparison to our results (there is no northwest–southeast tilt in
the northward vegetation push during AHPs). Nonetheless, the average evolution of total vegetation changes (one minus the
fraction of desert plus barren biomes) agrees well with the domain-average performance of our method, although the estimated
number of AHPs seems to be lower in their case.

Similarities with previous findings that include other models mean that the dominant pattern of variation we find for total
vegetation cover fraction may be model-independent and perhaps an adequate representation of a large-scale structure un-
derlying changes happening in North Africa during the last glacial cycles (including the AHPs as peak amplitude). In such
case the simple pattern scaling relationship established here could indeed approximate to some extent the land cover changes,
following mainly forcing values of orbital parameters, GHG levels and ice-sheet volume. The pattern scaling could be useful
to produce quick estimates of North African vegetation coverage, since simulating glacial cycles even at coarse resolution can
take excessively long computing times.

However, an important caveat about pattern scaling for vegetation fraction must be considered. Because this quantity is
bounded ($0 \, \%$ to $100 \, \%$), it can happen that in some cases the scaling "overshoots" the estimation of the changes, since there
is some maximum possible amount of change (from PI) leading to completely bare ($0 \, \%$) or vegetated ($100 \, \%$) grid cells.
Overshooting could happen, for instance, under more intense forcing (than in the last glacial cycle), or in case there is a greater
sensitivity to the forcing. This implies the linear scaling may work well for vegetation fraction in this study only because of a
dry bias in the model that prevents simulation of stronger (non-linear) greening.

Another factor we neglect in the scaling approach is the role of non-linear effects. Critical thresholds related to the climate
forcing may prevent the pattern scaling approach from correctly representing parts of the changes in the domain (Brovkin et al.,
1998; Claussen et al., 2003; Duque-Villegas et al., 2022), most likely near transition zones like around the southern fringe of
the Sahara or where there is steep topography like towards East Africa. Those are some of the places that stand out when
we assess the errors in the pattern scaling (see Fig. 5). Moreover, the dynamical systems that govern North African moisture
changes are connected via Walker and Hadley circulations with remote locations whose response to changes in the background
climate also could induce more or less greening during simulated AHPs (e.g. Kaboth-Bahr et al., 2021). In this case we have
not looked into such tele-connection mechanisms. Although it is possible that some of the other coherent modes of variance
improve our estimates (i.e. reduce emulation minus simulation differences), we opt for the simplicity of capturing only the
variance at the largest scales, at the cost of a mean error of about $\pm 10 \, \%$ vegetation fraction change.



In our simulations of millennial-scale future climate change, we see the contrast between GHG-enabled and orbital-driven vegetation patterns of AHPs. The vegetation pattern under high GHG forcing (Fig. 7e) is similar to 21st century projections of Sahel rainfall changes (e.g. Bony et al., 2013; Gaetani et al., 2017). In both cases (high GHG and 21st century projections)

the largest positive anomalies concentrate near central and eastern Sahel. However, in our case (high GHG) a key distinction is that the orbital configuration is changed considerably from present-day, thus we see also some vegetation growth in West Africa and approaching the Northern Tropic. Still, the western and northward development of vegetation under high GHG forcing is not as extensive as in the orbital-driven AHPs of the late Quaternary. This is related to the differences in dynamical (circulation) and thermodynamical (water vapour) responses to each of these forcing factors (Claussen et al., 2003; D'Agostino

et al., 2019). Such competing effects complicate AHP pattern predictions in future climates.

## 5   Conclusions

After simulating a realistic climate evolution for the last glacial cycle, we produce new estimates of the vegetation cover patterns during the last four AHPs. A comparison with available data sources from the geological record and relevant modelling studies indicates that the estimated patterns capture some of the continental-scale features in the evidence. Nevertheless, we identify a

"desert bias" for the model version at the resolution we use, which frames our results rather as probable minimum estimates. We reduce the high-dimensionality in the variability of the vegetation cover changes in North Africa to a single dominant pattern that accounts for $73\,\%$ of the variations. We limit our approach to a single dominant pattern because only for it we can establish a simple link to forcing variables. When we scale this pattern according to climate forcing variables we are able to replicate to an acceptable extent the simulated changes. Considering the high computational costs of running glacial cycle

simulations, we think our approach could be used as a quick diagnostic tool for palaeo-vegetation cover in North Africa, which we demonstrate by estimating vegetation cover changes for the last eight glacial cycles with considerable success. Moreover, extension of our simulation several millennia into the future reveals how different AHPs form under strong GHG forcing. Predicting future vegetation patterns should not be possible assuming a simple scaling of climate forcing.

*Code and data availability.*   Model source code of MPI-ESM is available to the scientific community upon request to MPI-M. Model output

data and post-processing Python scripts to reproduce numbers and figures in the manuscript are archived at https://doi.org/10.17617/3. HQTV1J (Duque-Villegas, 2024). The Lisiecki and Raymo (2005) data in Fig. 1d are available at http://lorraine-lisiecki.com/LR04stack.txt. The Deplazes et al. (2013) data in Fig. 1f are available at https://doi.org/10.1594/PANGAEA.815882. The Tierney et al. (2017a) data in Fig. 2a are available at https://doi.org/10.25921/k92a-8583. The Skonieczny et al. (2019) data in Fig. 2b are available as part of the supplementary material in the online publication. The Ehrmann and Schmiedl (2021) data in Fig. 2c are available at https://doi.org/10.1594/PANGAEA.

923491. Fig. 6b includes data from Loulergue et al. (2008), Lüthi et al. (2008) and Schilt et al. (2010) available at https://doi.org/10.25921/ gfsj-jc86, https://doi.org/10.25921/xgzs-gd10 and https://doi.org/10.25921/yxhy-3g37, respectively. The Emeis et al. (2000) data in Fig. 6d are available at https://doi.org/10.1594/PANGAEA.704622. The Grant et al. (2022) data in Fig. 6e are available as part of the supplementary





material in the online publication. The Krapp et al. (2021) data used for comparison can be found at https://doi.org/10.17605/OSF.IO/8N43X. The SPOT/PROBA-V/S3-OLCI and GPCP data in Fig. C1 were provided by the ICDC, CEN at Universität Hamburg.

**Appendix A: Backwards extension of transient simulation**

The complete 134-kyr simulation is mostly a single experiment that was run continuously from 125 ka to 0 ka. This interval is planned for studies about global climate of the last glacial cycle. However, because for this study we are particularly interested in the regional climate of North Africa, we choose to complement this long experiment with another shorter, independent, and overlapping experiment that has an initial date slightly earlier, spanning from 134 ka to 120 ka. We could use this to extend the longer experiment backwards in time. The main reason is to include in our analysis the tropical insolation maximum near 127 ka, which according to available evidence is an important time of extensive humidity over North Africa. We opt for such an extension due to the computational expense and long waiting time (around two years) that it would take to re-run an entirely new simulation. The overlap in the experiments is to measure their level of agreement near a merging point. They should not be drifting away from each other nor at completely different states.

We find the splicing together of both experiments to have only a small consequence, producing a low amplitude change that can be ignored for the study of the long-term climate signals we focus on for the study of AHPs. Figure A1 shows an example of the state evolution of both experiments for one grid cell in a highly variable region in the North Atlantic (Fig. A1a) and at the centre of our region of interest (Fig. A1b). The experiments are spliced together at 120 ka to produce the long simulation starting at 134 ka.

**Appendix B: Background climate near simulated AHPs**

To be able to compare more objectively the differences between simulated AHPs we include Fig. B1, where we show the global conditions of near-surface temperature and precipitation near the times of the AHPs. The pre-industrial (PI) conditions (Fig. B1a), near the year 1850 CE, show the typical poleward temperature gradient profile and the precipitation maximum around the equator. In Fig. B1b we see the climate during MIS 5e is overall much warmer than during PI, except around the Sahel in North Africa, where there is more transpiration because of the extended vegetation cover during the AHP. MIS 5c and MIS 5a show very similar conditions, also with a slightly warmer Arctic Ocean and Siberia regions. In great contrast, the Holocene (MIS 1) shows much colder temperatures over Greenland and the North Atlantic Ocean, due to an excessive amount of sea ice that remains in this region after a strong meltwater pulse near 14.4 ka. Because of this, the Holocene AHP we present is not as green as seen in previous applications of the same model at the same spatial resolution. The precipitation changes (Fig. B1c) show clearly the shift of moisture and ITCZ towards the Northern Hemisphere during the AHPs. The largest increments in precipitation happen not only in North Africa, but also over the Indian Ocean and over India. Precipitation changes during the Holocene AHP are much weaker than during the other AHPs.





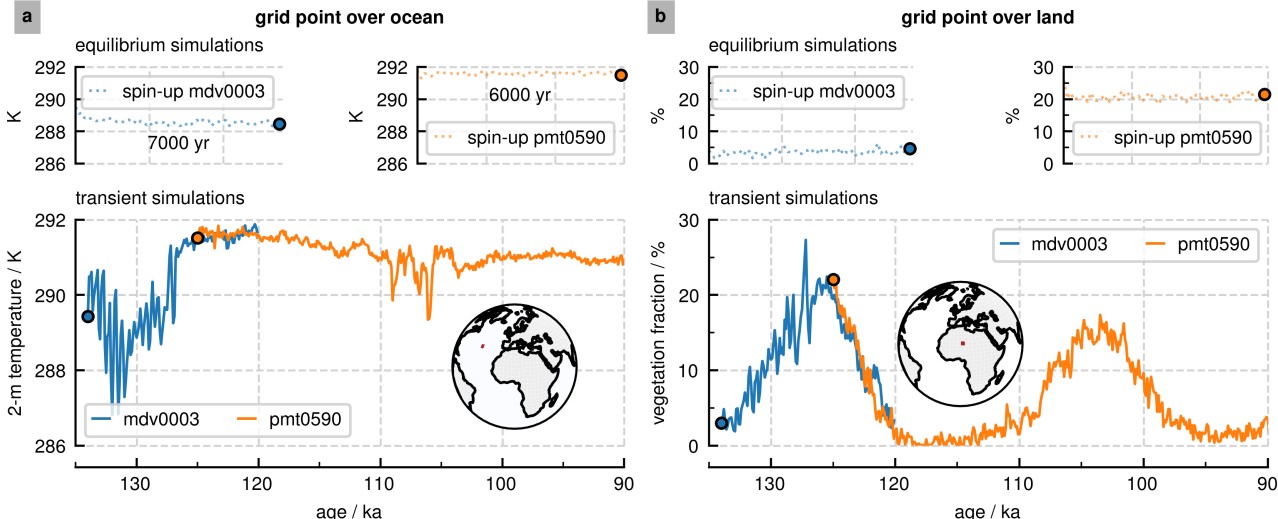

**Figure A1.** Extending the simulation deeper in time by merging overlapping experiments. Comparison of equilibrium and transient experiments for single grid cells (a) over the North Atlantic ($33.75°$ W $35.26°$ N) and (b) over land in North Africa ($7.5°$ E $20.41°$ N). Experiment "mdv0003" spans between $134\,\text{ka}$ to $120\,\text{ka}$, while experiment "pmt0590" from $125\,\text{ka}$ to $0\,\text{ka}$. We zoom time on the interval close to the merging point. Circle markers show the end state of the equilibrium experiments, also the starting point of the transient experiments.

## Appendix C: Comparison to present-day observations

For an assessment of the dry bias in this specific implementation of the MPI-ESM model, we also compared the simulated
present-day conditions of vegetation cover fractions and precipitation with some observational products in Fig. C1. In this case
"present-day" is defined as the climatological mean of 31 years of model output, corresponding to the interval from 1990 CE
to 2020 CE. The observational products were remapped to the model grid using a bicubic interpolation and averaged in time
to obtain climatologic values. It is important to note that the simulations did not include any anthropogenic land-use changes,
therefore, in many places (like near the equator) it is expected that the simulated vegetation cover is an overestimation (potential
vegetation cover). Figure C1a–c shows that the patterns of vegetation are similar, but the model underestimated the vegetation
cover around the Sahel (in spite of not including land use changes). A dry bias is clearly seen in the precipitation differences
in Fig. C1d–f, especially around the Sahel. The large positive anomalies at the equator (almost the only positive anomalies)
indicate the maximum of precipitation in the simulation may be displaced southward, relative to the observational product data.

*Author contributions.* MDV and MC developed the research idea. MDV and TK performed the climate simulations. MDV led the formal
analysis and drafted the manuscript. TK provided model code and input data. JB and CHK contributed to the methodology and formal
analysis. All authors participated in the analysis of results and manuscript composition.





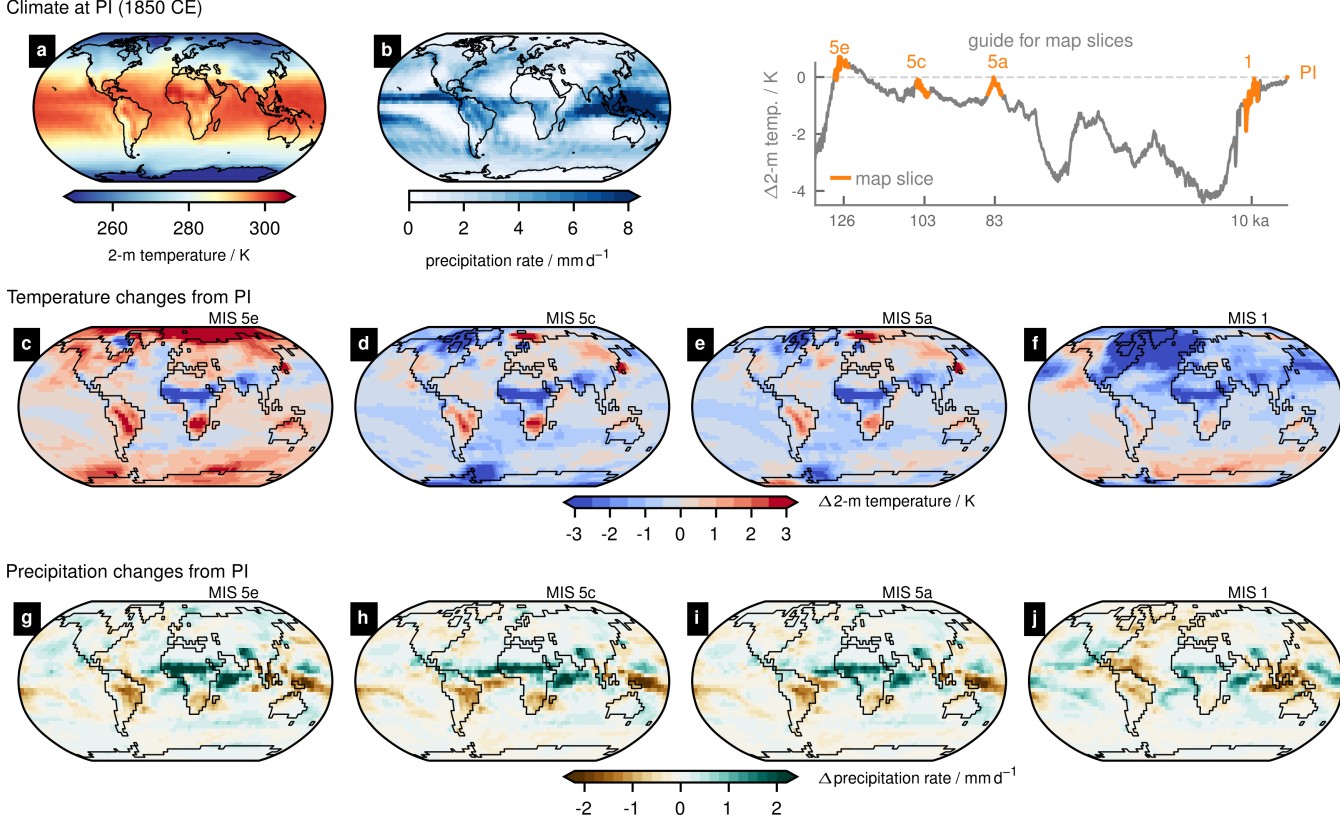

**Figure B1.** Background climates during AHPs: pre-industrial (PI) era means of (a) near-surface temperature and (b) daily precipitation rate fields; (c–f) temperature changes from PI; and (g–j) precipitation rate changes from PI. Modern coastlines are shown for PI, while AHP time slices show the coastlines according to sea level changes. An inset axes with global mean surface temperature highlights the AHP time slices.

*Competing interests.* At least one of the (co-)authors is a member of the editorial board of *Climate of the Past*. The authors have no other competing interests to declare.

*Acknowledgements.* We thank Anne Dallmeyer (MPI-M) for valuable input about an earlier version of the manuscript. Simulations are part
of the "PalMod" project, funded by the German Federal Ministry of Education and Research (BMBF) through the Research for Sustainability (FONA) initiative (grant no. 01LP1921A). This work also contributes to the project "African and Asian Monsoon Margins" of the Cluster of Excellence EXC 2037: Climate, Climatic Change, and Society (CLICCS). Thanks to ICDC, CEN, University of Hamburg for data support to produce Fig. C1. We acknowledge the support of the German Climate Computing Center (DKRZ) in providing computing resources and assistance. Analyses and figures were produced using Python, including libraries NumPy, Matplotlib, cartopy, xarray, SciPy, pandas,
statsmodels, and cftime.



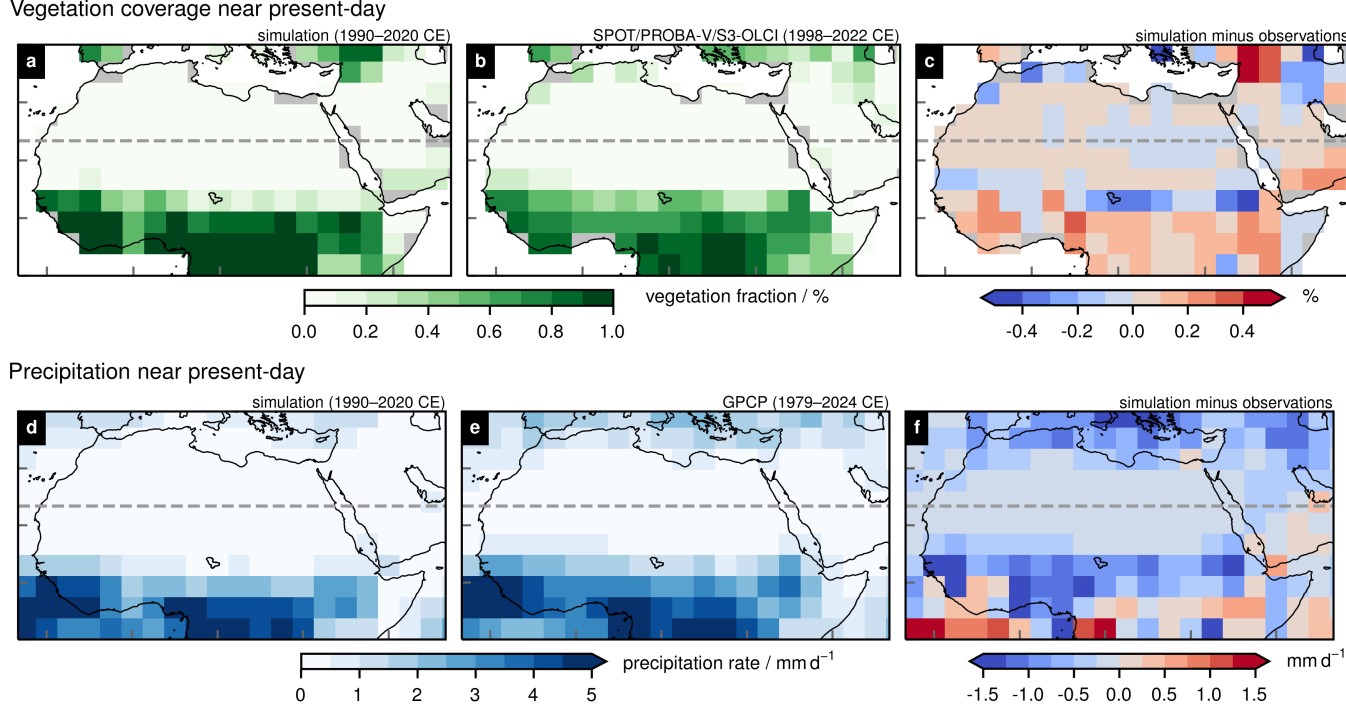

**Figure C1.** Comparison in North Africa of simulated present-day (1990–2020 CE) conditions to observational products for: (a–c) vegetation cover fractions and (d–f) daily precipitation rates. (b) vegetation cover data are from SPOT/PROBA-V and PROBA-V/S3-OLCI products of the Land Service of Copernicus (Baret et al., 2013; Camacho et al., 2013). (e) precipitation data are from the Global Precipitation Climatology Project (GPCP; Adler et al., 2016).

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
