# Peer review of "Pattern scaling of simulated vegetation change in North Africa during glacial cycles"

_Climate of the Past, 2024_

## Author Response (AR1)

**Author's Response on CP-2024-61**

Mateo Duque-Villegas (on behalf of all co-authors)
*mateo.duque@mpimet.mpg.de*

4th February 2025

We thank both reviewers, editor Manuel Chevalier and the Copernicus Editorial Team for their help with improving our manuscript. Below we compile all reviewers' comments in blue, our responses in black and **manuscript modifications in bold red**. In the modifications we refer to the line numbers in the *revised manuscript*.

**Referee Comment 1 (RC1) by anonymous referee**

[0] The paper from Mateo Duque-Villegas et al. uses MPI-ESM v1.2 for the past 134 ka to simulate northern African hydroclimate and vegetation to reconstruct past AHPs. It aims to construct a pattern scale model to estimate AHPs of the past 800 ka. The model uses forcing variables (insolation, ice volume, and GHG) and is tested against proxy variables such as d18O (vs. 2K T), dP (vs. L*), and vegetation (vs. isotopic depletion and dust). Then, past AHPs are defined as peaks in the pan-Saharan vegetation coverage (%) and analyzed spatially. Here, EOFs capture dominant patterns of vegetation change, which are used to fit a linear model between the forcing variables and the PC1. They state orbitals influence around 66% of the linear model outcome. Next, the pattern scaling is applied and compared against MPI-ESM output, showing a trustworthy approach and extending the temporal extent back to 800 ka. The comparison with the Saporpel record from the Mediterranean does show a good agreement when using some arbitrary thresholding. Last, the pattern scaling approach and their MPI-ESP are used for future SSP scenarios with moderate and intensive GHG emissions for the coming ten kyr. Differences are observed, mainly when the pattern scaling linear model is used for the intense GHG emissions, as this quasi-empirical model has no data example for that.

[1] The paper presents an insightful, well-written approach and uses well-curated figures to show its results. The paper structure is sometimes unusual with literature work and comparisons and discussions within the result section and, therefore, a short discussion section due to the linear progressing type of analysis. I like it, but others can see this differently. The length and the grade of detail are sufficient and allow fluent reading.

[2] The method and the presented results seem logical and reasonable; no significant flaws were detected by my side, even though I am not an expert in GCMs, especially not for this model. Therefore, from my modest perspective, I see the paper as acceptable, but there are some suggestions and comments from my side:

**Major comments**

[3] The temporal framework in which the model results are presented is based on glacial cycles and the marine isotope stages, whereas the research target is mainly tropical. There are arguments to see this as problematic, as it can be a north-western perspective that acts as a framework to understand tropical climate, leading to false reasoning about forcing factors. They may also lead to false conclusions when they are used to discussing tropical climates if NH climate implications are applied. In contrast, cycles of the Monsoon Index could simply be counted and labeled, or absolute ages could be used.

Indeed, the focus of our study are the tropics, but our simulations are global, and we discuss the climate and vegetation changes in northern Africa in a global context. We find the MIS terminology useful, because it characterizes the (background) state of the global climate system while the changes in the Sahara occurred. Many of the proxy data publications of late Quaternary climate change in northern Africa also include such MIS referencing (e.g. Ehrmann et al., 2017; Grant et al., 2022; Rossignol-Strick, 1983; Skonieczny et al., 2019; Tierney et al., 2017). Alternative labelling such as AHP1, AHP2, etc., or AHP of Monsoon Index (MI) 1, AHP of MI2, etc. would not provide such a global context. We would be surprised, if the MIS terminology would lead to false reasoning about the forcing factors. We clearly specify the forcing factors in terms of the (tropical) Monsoon Index for scaling (although the numerical simulations are driven by the global changes in the meridional insolation), the radiative forcing changes linked to GHG changes, and the extent of Northern Hemisphere ice sheets (although the prescribed Antarctic ice sheet also changes slightly in the numerical simulations). Nevertheless, we agree that the absolute ages should be clearly stated. In a revised version of the manuscript, we will explain our use of MIS labels to identify the four AHPs, as well as make sure absolute ages accompany their first appearance in the text.

**We added a new sentence in lines 111–112, briefly explaining our use of MIS labels. We added also approximate ages for MIS 5c and 5a in line 177, and for MIS 1 in line 213.**

[4] The draft presents its results mainly as a cost-efficient estimation of past AHP patterns based on the simple linear fit. However, the fit itself is of interest, as it shows that orbitals are the main drivers of AHPs, not GHGs or ice sheets. Why is this not presented as an exciting result to be discussed?

We agree with the reviewer that our result regarding the dominance of orbital drivers is exciting. We published this result already in a previous paper (Duque-Villegas et al., 2022), but we are happy to re-iterate that also the more complex model exhibits this dominance in the forcing factors (previously we used an intermediate complexity model). In a revised version of the manuscript, we will explain in the discussion section how both studies are connected by this finding.

**We added a new paragraph in lines 388–395, highlighting this connection between studies.**

[5] Vegetation and vegetation feedback are non-linear and have multimodal distributions. However, EOF assumes unimodality and works better with a Gaussian distribution. How is this considered?

We discuss the linear limitation of our scaling approach in the discussion section (lines 392 ff.). Indeed, such non-linear climate-vegetation interactions appear in the simulations with the numerical climate model, as we show for the Sahara in Fig. 2. However, the feedback is not as strong as seen in our previous application with an intermediate complexity model, and we suspect part of the success of our EOF method could be attributed to this rather weak feedback. Moreover, the problem of spatial heterogeneity in the local vegetation responses clearly limits the applicability of our uni-modal EOF approach, yet we argue the method should be capturing a common general trend of change imposed by the forcing factors we consider, all other things being equal. At least at our working horizontal resolution (grid spacing of about 400 km) we find such an uni-modal response that explains much of the regional variability along glacial cycles. We will emphasize these issues in a revised version of the manuscript.

**We added a short sentence in line 418 explicitly justifying the uni-modal approach.**

**Minor comments**

[6] L5: Capitalize African Humid Period (AHP)?

In a revised version we will capitalize the AHP.

**We capitalized AHP in lines 2 and 19.**

[7] L25: „1.7 ka to 4.2 ka,“ Where are these definite numbers from? There is a reference needed. Also, a broadened perspective on when, where, and how the AHP ends would be helpful (e.g., abrupt/ gradual, eastern vs. western Africa, or N-S transect, e.g., Shanahan et al., 2015)

The missing reference is Claussen et al. (2017), but we agree that with the broadened perspective we do not need such precision in the ages. In the revised version, we will briefly mention the spatial heterogeneity and abrupt and gradual changes making also a reference to Dallmeyer et al. (2020).

**We added the reference in line 26 and a new sentence in lines 29–32 making mention of the spatial complexity.**

[8] L110: Marine sediment reflectance is rapidly introduced here and would need some more explanation for the non-marine audience

The revised version of the manuscript will include a sentence in the main text next to the citation of the Cariaco Basin record.

**We added a new sentence in lines 149–151, giving more context about the proxy.**

[9] L115: More wording could be helpful to explain that the output of the model is compared to proxy results and other model insights

The revised manuscript will include an opening paragraph to introduce in such way the results section.

**We added a new paragraph in lines 122–126 that introduces the results section.**

[10] L145: Eastern Africa instead of East Africa, as East Africa refers to the colonial name, whereas eastern Africa is the geographical term

We thank the reviewer for the clarification. This will be corrected in the text.

**We corrected this in the caption of Fig. 2 and lines 157, 233, and 412.**

[11] L165: If the last AHP, which has the best ground truth so far, is not correctly reconstructed, how do we assume the model works correctly?

The climate model we use has similar assumptions and biases as other state-of-the-art general circulation models (GCMs). It is periodically evaluated as part of model inter-comparison palaeo-experiments (Braconnot et al., 2012; Brierley et al., 2020). Recent evaluations of the simulated Holocene show the model aligns well with general trends in current proxies for northern Africa and globally (Dallmeyer et al., 2020, 2021). Combined with several deglaciation experiments since the Last Glacial Maximum until present, which were performed with the same model (e.g. Dallmeyer et al., 2022; Kleinen et al., 2023), it gives us confidence the model tends to work reasonably well, at least in broad agreement with current evidence, although with the known limitations of GCMs. It was those previous applications that alerted us that the Holocene AHP should not have appeared as weak as it did. The extent to which a weak Holocene AHP affects our analyses and the pattern-scaling is difficult to measure, but we do not expect that our results would be qualitatively different.

**We added a new sentence about this in line 369.**

[12] L235: I think this is an exciting output of the study as it shows, with a simple linear model, the contributions of the forcing factors to Africa's climate heartbeat. For the last sentence, I would pronounce the weakening of the orbital forcing without necessarily increasing NH forcing on tropical African climate.

We agree with the reviewer on rather emphasizing the role of the weakening of the orbital forcing in the last sentence.

**We modified the sentence accordingly in line 251.**

[13] L325: Indeed, pattern scaling is an empirical method, and there is no precursor to having the model trained on, so it is extrapolating; hence, reaction patterns to this GHG forcing are simply unknown.

We thank the reviewer for the clarification. In spite of the uncertainty in such extrapolation we find it interesting to discuss jointly the output of the dynamical numerical model and the pattern scaling in such future scenarios, as a way to understand the differences between changes driven mainly by Earth's orbit or GHGs.

**We adjusted our wording in line 351 to reflect this.**

[14] L375: An AHP SW-NE tilt exists in the Krapp et al. (2021) dataset, foremost in the MAP. It is weak and underestimated compared to the terrestrial observations, but it exists. We thank the reviewer for pointing this out, since we had only looked at the biomes output. We will revise the comparison with this dataset.

**We removed our wrong assertion in line 400.**

**Referee Comment 2 (RC2) by Shivangi Tiwari**

**General comments**

[0] In my view, this study is very interesting and relevant, and quite novel. Barring some grammatical problems, the manuscript is well-written in terms of coherence and flow and the figures are well-made. In my view, it is certainly a study which should be published and would be of interest to the paleoclimate community (especially AHP workers). However, it requires some revisions before publication. There are two major issues that the authors should address:

[1] 1. The under-estimation of the Holocene AHP is critical. This is not just because the model may be under-representing AHPs in general, because that would hopefully apply to all simulated AHPs and hence not greatly affect the linear scaling. The issue is that the proposed mechanism (unexpected MOC variability) for this under-estimation appears to apply only to the Holocene AHP. It appears to me that similar MOC variability also occurred before the AHP in MIS5e. Does it lead to a similar under-representation of that AHP? In any case, either one or both of the interglacial AHPs are under-estimated but the other two are not. Can a linear scaling model really be applied in this case?

To clarify: my issue is not with the scaling of the orbital forcing, which is evident in Fig. 1a. Intuitively, this would likely lead to a scaling of the AHP representation. My issue is with the ability of your simulations to show this scaling given the imbalanced representation of the different AHPs.

We agree with Dr. Tiwari that the under-estimation of the Holocene AHP is important, however, we do not expect that it alone can upset greatly the relationship we find in concert with the other AHPs. Since orbital changes appear to be the dominant forcing of AHPs in the past, the limitation that other processes/forcing factors operate in some AHPs (e.g. AMOC variations), but not all, is tolerable for applying a regression model. It certainly has an impact on how well the individual AHPs are represented by the regression model. It is important that the influence of the orbital boundary conditions is clearly visible in all AHPs and is not masked by another process. Nonetheless, we are confident that only the Holocene AHP was affected by some odd MOC variability that appeared after the strong Meltwater Pulse 1a (MWP1a) near 14.4 ka. The MOC variability near the AHP of MIS 5e is not as great (away from 0 Sv in Fig. 1e) and is in line with the discharges of the ice-sheet reconstruction. Compared to our previous other (shorter) simulations, only the Holocene AHP appears disturbed, while the AHPs of MIS 5 appear consistently as in previous simulations. Therefore, even when the model is under-representing AHPs in general (dry bias, line 390 ff.) we believe the linear relationship we establish should remain valid. In a revised version of the manuscript we will mention in the main text parts of Appendix B that refer to the previous experiments that make it clear only the Holocene AHP was unexpectedly weakened by AMOC variations.

**We added a new sentence in line 181 to explain why MIS 5e is *expectedly* affected by AMOC.**

[2] 2. To me, a study of the future where ice-sheets are kept unchanged is meaningless, especially when ice volume is one of three predictors. I'm afraid the point of Section 3.6 is quite lost on me. From my viewpoint, it would be a perfectly good and "complete" study without attempting to simulate the future climate (imperfectly) as well.

We agree with Dr. Tiwari that the assumption of unchanged ice sheets over the next 10 kyr is a bold one. Indeed, ice volume as a predictor in our method is mainly important for past glaciation cycles, while for the two future scenarios it remains neutral. However, we do not think that it is meaningless. Specifically, we refer to the study by Ganopolski et al. (2016), where they suggest that only in the case of decreasing $CO_2$ concentration we can expect some increase in the ice sheet extent. In other cases, their simulations suggest a nearly constant or smaller ice extent. In a revised version of our manuscript, we will emphasize that our assumption of a constant ice sheet extent over the next 10 kyr is only a very first approximation. We suppose that this assumption will only marginally affect our conclusion that the possible future patterns of Saharan greening do not look like the patterns of the last 130 kyr.

**We added a new sentence in line 114 mentioning our approximation.**

[3] Another relatively minor issue is that I would have liked to see more evidence in favor of their scaling approach for ancient AHPs through more extensive comparison with other proxy records or modelling studies. This is discussed briefly, but to me, elaborating this would be most beneficial in reinforcing the applicability of this technique to ancient AHPs.

We agree that it is key to evaluate our simulations and pattern scaling method with available sources. We compare our findings with some marine proxies and a few other modelling results, but we keep it brief because we do not think that a "more extensive comparison" is possible yet. Only for the Holocene AHP more extensive comparisons have been made and we refer to some of them which are linked to our climate model (e.g. Dallmeyer et al., 2020), but for the AHPs of MIS 5 the proxy records are too few and fragmented, and therefore evaluation or inter-comparison of climate models is still work in progress (e.g. Otto-Bliesner et al., 2021).

**Paragraph in lines 381–387 clarifies difficulties in comparison with records.**

**Specific comments**

[4] L18: Perhaps deMenocal et al. (2000) is not the best reference here, because you mention water bodies and expansion of vegetation cover only.

In a revised version of the manuscript we will mention that such covers modulated dust output to the Atlantic Ocean.

**Modified accordingly in line 18.**

[5] L22: I don't think our understanding of the extent and timing of the AHP changes depends on climate simulations (and their discrepancies with records), but instead, it is more to do with the proxy data limitations is well taken. You could highlight the point about the discrepancies in a separate sentence.

We agree with the reviewer that the sentence lacks clarity.

**Modified accordingly the sentence in lines 23–24.**

[6] L25: The references Blanchet et al., 2021 and Ehrmann et al., 2017 are not really wrong, but perhaps misplaced here.

We agree they can be omitted here.

**Removed the misplaced references.**

[7] L32: You should either provide a reference for the second part of the sentence which focuses on climate models, or break this sentence into two and present the second part as a general point.

We will refer to Tierney et al. (2020) where they mention computational expenses in palaeo-climatology.

**Added reference in line 36.**

[8] L54: Both the references are related to the Holocene AHP only, hence misplaced in a paragraph that focuses on ancient AHPs. You could cite the same papers but to say that this is a problem with the Holocene AHP simulations which may extend to the ancient AHPs too (depends on the point you want to make, of course).

We agree it is important to make this distinction in a revised version.

**Adjusted the sentence and removed references in lines 57–58.**

[9] L109: Keeping ice sheets unchanged at present day levels is a critical limitation, in my opinion.

As discussed in a previous point, we will refer to the study by Ganopolski et al. (2016).

**We added a new sentence in line 114.**

[10] L138: This is confusing. What is the simulation offset from: the proxy record or the orbital monsoon index? It looks like both to me (at different times). In Fig. 1f, the model precipitation shows more variability and the proxy record is smoother. I see from the figure caption that the proxy record is smoothened, but it would be helpful to mention that here too.

We thank the reviewer and we will clarify this in a revised manuscript.

**We modified the sentences in lines 151–153 for clarity.**

[11] L157: Is this lag not related to the time taken for vegetation to develop and fully expand over the region?

This is also likely a cause for a lag, but only of a few centuries. In a revised version of the manuscript we will explain it is *most of the* lag what is related to the orbital configuration.

**We modified accordingly the sentence starting on line 171.**

[12] L164: Wouldn't the effect of the MOC variability apply to MIS 5e too?

Yes, but the difference is that the MOC variability during MIS 5e is expected because it is part of the meltwater discharges related to the ice-sheets reconstruction, while in the case of the Holocene it is unexpected because the meltwater discharges are not so large that they should cause such AMOC variations (see AMOC in Kleinen et al. (2023)).

**We added a new sentence lines 181–182.**

[13] L303: I think it is key to quantify how many of the simulated AHPs do agree with sediment core signals, since that would be the most robust evidence in favour of applying this linear scaling to ancient AHPs.

We will add this number in a revised version of the manuscript.

**We added the number on line 316.**

[14] L327: I think this is a crucial limitation of applying your linear scaling model to the future. Simulating the future with invariant ice-sheets is both unrealistic and meaningless. It is additionally problematic if your linear scaling model has an ice volume parameter. For me, the Section 3.6 does not hold any meaningful results and I do not see the point of including it in the manuscript. (Perhaps the significance has eluded me, and the authors could explain it better.)

We addressed this point in a previous comment. We think it is meaningful to contrast Sahara greening patterns according to different dominant drivers (orbit vs. GHGs), at least as a very first approximation.

**No action.**

[15] L335: Independently of the previous comment, I would argue that it is not correct to say that the simulated pattern "does not resemble any of its AHP predecessors". The absence of the northward vegetation expansion along the west coast is strange, but hardly the only criterion to be employed. (I see a decent zonally-extended expansion and also a meridional pattern in eastern Africa, for example.)

We thank the reviewer and we agree this should be fixed in a revised version of the manuscript.

**We adjusted the sentence in line 348 accordingly.**

[16] L336: I don't think either of the papers cited is appropriate for a comparison (or for showing agreement) here. Neither discusses the time after 9 kyr. Pausata et al. (2020) discuss the future in terms of geoengineering projects only. If you would like to keep this statement, please clarify how those studies have results similar to the differences you're talking about.

We thank the reviewer for pointing this out. Indeed Pausata et al. (2020) is misplaced here, and it should have been Gaetani et al. (2017). The relevance of D'Agostino et al. (2019) and Gaetani et al. (2017) is because both studies look at the response of the West African Monsoon under high GHGs scenarios. Although the orbital configuration at 9 kyr AP is different in our experiments (and explains part of the Saharan greening), we argue that the effects of the high GHGs scenarios is comparable across these studies and ours.

**We adjusted the references and wording in lines 349–351.**

[17] L337: I see that this is the same point I was trying to convey earlier. However, I don't think it really qualifies to be a "result" of this study, but is instead a foreseeable limitation. In a revised version we will rephrase this to clarify that this is an expected limitation of our scaling approach because the fitting data did not include such ranges of GHGs because they did not happen during the late Quaternary.

**We adjusted the sentence in line 351 to explain how this is a limitation by design.**

[18] L369-370: This line is unclear. How are you comparing vegetation changes with precipitation changes from a model-data comparison? This is the first mention of any comparison with Scussolini et al. (2019). The results from the comparison should be mentioned somewhere in the Results section before. In a revised version we will clarify the qualitative nature of this comparison and its limitations.

**A new paragraph in lines 381–387 clarifies this brief comparison.**

**Technical corrections**

We appreciate all the suggestions and will address them in a revised version of the manuscript.

**All suggestions were addressed in various places in the text.**

**References**

Braconnot, P., Harrison, S. P., Kageyama, M., Bartlein, P. J., Masson-Delmotte, V., Abe-Ouchi, A., Otto-Bliesner, B. L., & Zhao, Y. (2012). Evaluation of climate models using palaeoclimatic data. *Nature Climate Change*, *2*(6), 417–424.

Brierley, C., Zhao, A., Harrison, S. P., Braconnot, P., Williams, C. J. R., Thornalley, D. J. R., Shi, X., Peterschmitt, J.-Y., Ohgaito, R., Kaufman, D. S., Kageyama, M., Hargreaves, J. C., Erb, M. P., Emile-Geay, J., D'Agostino, R., Chandan, D., Carré, M., Bartlein, P. J., Zheng, W., ... Abe-Ouchi, A. (2020). Large-scale features and evaluation of the PMIP4-CMIP6 midHolocene simulations. *Climate of the Past*, *16*(5), 1847–1872.

Claussen, M., Dallmeyer, A., & Bader, J. (2017). Theory and modeling of the African Humid Period and the Green Sahara. In *Oxford research encyclopedia of climate science* (pp. 1–40). Oxford University Press.

D'Agostino, R., Bader, J., Bordoni, S., Ferreira, D., & Jungclaus, J. (2019). Northern Hemisphere monsoon response to mid-Holocene orbital forcing and greenhouse gas-induced global warming. *Geophysical Research Letters*, *46*(3), 1591–1601.

Dallmeyer, A., Claussen, M., Lorenz, S. J., & Shanahan, T. M. (2020). The end of the African Humid Period as seen by a transient comprehensive Earth system model simulation of the last 8000 years. *Climate of the Past*, *16*(1), 117–140.

Dallmeyer, A., Claussen, M., Lorenz, S. J., Sigl, M., Toohey, M., & Herzschuh, U. (2021). Holocene vegetation transitions and their climatic drivers in MPI-ESM1.2. *Climate of the Past*, *17*(6), 2481–2513.

Dallmeyer, A., Kleinen, T., Claussen, M., Weitzel, N., Cao, X., & Herzschuh, U. (2022). The deglacial forest conundrum. *Nature Communications*, *13*(6035), 1–10.

Duque-Villegas, M., Claussen, M., Brovkin, V., & Kleinen, T. (2022). Effects of orbital forcing, greenhouse gases and ice sheets on Saharan greening in past and future multi-millennia. *Climate of the Past*, *18*(8), 1897–1914.

Ehrmann, W., Schmiedl, G., Beuscher, S., & Krüger, S. (2017). Intensity of African humid periods estimated from Saharan dust fluxes. *PloS ONE*, *12*(1, e0170989), 1–18.

Gaetani, M., Flamant, C., Bastin, S., Janicot, S., Lavaysse, C., Hourdin, F., Braconnot, P., & Bony, S. (2017). West African monsoon dynamics and precipitation: The competition between global SST warming and CO2 increase in CMIP5 idealized simulations. *Climate Dynamics*, *48*(3-4), 1353–1373.

Ganopolski, A., Winkelmann, R., & Schellnhuber, H. J. (2016). Critical insolation–$CO_2$ relation for diagnosing past and future glacial inception. *Nature*, *529*(7585), 200–203.

Grant, K. M., Amarathunga, U., Amies, J. D., Hu, P., Qian, Y., Penny, T., Rodriguez-Sanz, L., Zhao, X., Heslop, D., Liebrand, D., Hennekam, R., Westerhold, T., Gilmore, S., Lourens, L. J., Roberts, A. P., & Rohling, E. J. (2022). Organic carbon burial in Mediterranean sapropels intensified during Green Sahara Periods since 3.2 Myr ago. *Communications Earth & Environment*, *3*(11), 1–9.

Kleinen, T., Gromov, S., Steil, B., & Brovkin, V. (2023). Atmospheric methane since the last glacial maximum was driven by wetland sources. *Climate of the Past*, *19*(5), 1081–1099.

Otto-Bliesner, B. L., Brady, E. C., Zhao, A., Brierley, C. M., Axford, Y., Capron, E., Govin, A., Hoffman, J. S., Isaacs, E., Kageyama, M., Scussolini, P., Tzedakis, P. C., Williams, C. J. R., Wolff, E., Abe-Ouchi, A., Braconnot, P., Buarque, S. R., Cao, J., de Vernal, A., ... Zheng, W. (2021). Large-scale features of Last Interglacial climate: Results from evaluating the lig127k simulations for the Coupled Model Intercomparison Project (CMIP6) – Paleoclimate Modeling Intercomparison Project (PMIP4). *Climate of the Past*, *17*(1), 63–94.

Pausata, F. S. R., Gaetani, M., Messori, G., Berg, A., de Souza, D. M., Sage, R. F., & deMenocal, P. B. (2020). The greening of the Sahara: Past changes and future implications. *One Earth*, *2*(3), 235–250.

Rossignol-Strick, M. (1983). African monsoons, an immediate climate response to orbital insolation. *Nature*, *304*, 46–49.

Skonieczny, C., McGee, D., Winckler, G., Bory, A., Bradtmiller, L. I., Kinsley, C. W., Polissar, P. J., Pol-Holz, R. D., Rossignol, L., & Malaizé, B. (2019). Monsoon-driven Saharan dust variability over the past 240,000 years. *Science Advances*, *5*(1, eaav1887), 1–8.

Tierney, J. E., deMenocal, P. B., & Zander, P. D. (2017). A climatic context for the out-of-Africa migration. *Geology*, *45*(11), 1023–1026.

Tierney, J. E., Poulsen, C. J., Montañez, I. P., Bhattacharya, T., Feng, R., Ford, H. L., Hönisch, B., Inglis, G. N., Petersen, S. V., Sagoo, N., Tabor, C. R., Thirumalai, K., Zhu, J., Burls, N. J., Foster, G. L., Goddéris, Y., Huber, B. T., Ivany, L. C., Turner, S. K., ... Zhang, Y. G. (2020). Past climates inform our future. *Science*, *370*(6517, eaay3701), 1–9.